# Curcumae Radix Decreases Neurodegenerative Markers through Glycolysis Decrease and TCA Cycle Activation

**DOI:** 10.3390/nu14081587

**Published:** 2022-04-11

**Authors:** Seong-Lae Jo, Hyun Yang, Sang R. Lee, Jun H. Heo, Hye-Won Lee, Eui-Ju Hong

**Affiliations:** 1College of Veterinary Medicine, Chungnam National University, Daejeon 34134, Korea; jsr7093@nate.com (S.-L.J.); srlee5@naver.com (S.R.L.); heojh0624@naver.com (J.H.H.); 2KM Convergence Research Division, Korea Institute of Oriental Medicine, Daejeon 34054, Korea; hyunyang@kiom.re.kr

**Keywords:** Curcumae radix, Alzheimer’s disease, neurodegenerative disease, glycolysis, TCA cycle

## Abstract

Neurodegenerative diseases (ND) are being increasingly studied owing to the increasing proportion of the aging population. Several potential compounds are examined to prevent neurodegenerative diseases, including Curcumae radix, which is known to be beneficial for inflammatory conditions, metabolic syndrome, and various types of pain. However, it is not well studied, and its influence on energy metabolism in ND is unclear. We focused on the relationship between ND and energy metabolism using Curcumae radix extract (CRE) in cells and animal models. We monitored neurodegenerative markers and metabolic indicators using Western blotting and qRT-PCR and then assessed cellular glycolysis and metabolic flux assays. The levels of Alzheimer’s disease-related markers in mouse brains were reduced after treatment with the CRE. We confirmed that neurodegenerative markers decreased in the cerebrum and brain tumor cells following low endoplasmic reticulum (ER) stress markers. Furthermore, glycolysis related genes and the extracellular acidification rate decreased after treatment with the CRE. Interestingly, we found that the CRE exposed mouse brain and cells had increased mitochondrial Tricarboxylic acid (TCA) cycle and Oxidative phosphorylation (OXPHOS) related genes in the CRE group. Curcumae radix may act as a metabolic modulator of brain health and help treat and prevent ND involving mitochondrial dysfunction.

## 1. Introduction

With an increasingly aging population, the incidence of neurodegenerative diseases (ND) has rapidly increased in contemporary society. ND refers to the loss of the intrinsic function of nerve cells in the central nervous system and includes diseases such as Alzheimer’s and Parkinson’s disease. These are related to the accumulation of beta-amyloid (Aβ), alpha-synuclein (α-syn), and tau proteins, which damage the function of the nerve cells [1]. Aβ and tau proteins have a key role in Alzheimer’s disease. Amyloid precursor protein (App) transforms into Aβ through ꞵ-secretase and γ-secretase enzymes, and Aβ induces the apoptosis of neuronal cells through aggregation with the beta sheet [2]. Although tau protein maintains the structural stability of the microtubule, its overexpression or abnormal phosphorylation can cause Alzheimer’s disease via microtubule dysfunction and degradation [3]. Another brain disorder, Parkinson’s disease, is associated with decreased dopamine secretion and the excessive accumulation of α-syn aggregates [4]. Therefore, many new therapeutic strategies are suggested to reduce the accumulation of these proteins [5].

Metabolic therapy is one of the new therapies strategies studied recently in ND [6,7]. Given that the brain uses glucose as the main energy source, metabolically, glucose enters the brain through the blood–brain barrier [8], and glucose is oxidized to produce adenosine triphosphate (ATP) by glycolysis [9]. ATP plays a central role in cerebral bioenergetics, brain function, and ND. While glucose metabolism worsens in the brain, this dysfunction is aggravated in ND, such as Alzheimer’s disease and Parkinson’s disease [7]. Based on this evidence, we focused on therapy strategies centered on energy metabolism.

Curcumae radix is the dry root of Curcuma longa L., which is a plant affiliated with the Zingiberaceae that is used in traditional medicine [10,11]. Curcumae radix extract (CRE) contains three reference compounds: curcumin, demethoxycurcumin, and bisdemethoxycurcumin. Curcumae radix is beneficial in treating inflammatory conditions, metabolic syndrome, and various types of pain. Its main ingredient is curcumin, which has anti-inflammatory and antioxidant effects that inhibit the formation of reactive oxygen species (ROS) and inflammatory enzymes [12]. As demonstrated in several studies, the effects of curcumin are linked to its anti-inflammatory [13,14], antimicrobial [15], and anticancer properties [10,16,17,18] and, in the brain, curcumin is known to inhibit endoplasmic reticulum stress (ER stress) and apoptosis [19,20,21,22].

Previous studies found that curcumin stimulates glucose uptake and activates glucose metabolism in skeletal muscles [23,24]. However, it also reduces the expression levels of glycolysis-related enzymes and lactate dehydrogenase A in the liver [25,26,27,28]. In a diabetes study, curcumin inhibited adipose STAT3 activation, which is essential for signaling in insulin resistance onset [29,30]. Moreover, recent studies reported that curcumin affected protection against obesity and gestational diabetes [31,32]. Surprisingly, when treated with Curcumae radix, glycolysis metabolism and neurodegenerative markers decreased in the mouse brain and ND model-tau overexpression mouse brain. Though CRE is extensively studied, its role in energy metabolism in ND remains unclear. In this study, we focus on the effects of CRE on ND and energy metabolism.

## 2. Materials and Methods

### 2.1. Preparation of Curcumae Radix Extract (CRE)

Curcumae radix was obtained from Beneherb Agricultural Co. Ltd. (Jeju-si, Korea). The material was washed and rinsed with distilled water. These materials were prepared using a grinder–mixer. The origin plant samples were deposited in the Korea Institute of Oriental Medicine (KIOM) in South Korea (voucher specimen KIOM M 130110). Dried Curcumae radix power was extracted in 70% (*v*/*v*) ethanol for 120 min with sonication twice. The extracted 70% ethanol solution was filtered (Whatman No. 2) and then concentrated by a vacuum rotary evaporator (Büchi; Flawil, Switzerland) at 40 °C. To obtain Curcumae radix powder, the final extract was lyophilized (IlShin; Siheung-si, Korea) using a freeze-dryer (final yield of 11.4%, Appendix A). The compounds of Curcumae radix quantitative analysis were shown in a previous study [10]. 

### 2.2. Animals and Treatment

Mice on a C57BL/6N background were obtained from Orient Bio (Daejeon, Korea) and housed in a pathogen-free facility at Chungnam National University under a standard 12 h light:12 h dark cycle and fed standard chow with water provided ad libitum. All mouse experiments were approved and performed in accordance with the Chungnam Facility Animal Care Committee (202006-CNU-105). The mice used were 12 weeks old, and mice had a 2 week acclimation period. For 23 days, 5 times a week, they were injected orally, and for the last 7 consecutive days of administration (50 mg/kg body weight), the vehicle group was treated with the same capacity of distilled water as the CRE treatment. The mice were sacrificed at 30 days, and the organs were isolated. Six mice were used for each vehicle and CRE group in the experiment. C57BL/6-Tg (NSE-hTau23) Korl mice were obtained from the National Institute of Food and Drug Safety Evaluation (NIFDS, Cheongju, Korea) and housed in a pathogen-free facility at Chungnam National University under a standard 12 h light:12 h dark cycle and fed standard chow with water provided ad libitum. All mouse experiments were approved and performed in accordance with the Chungnam Facility Animal Care Committee. The mice used were 24 weeks old, and mice were administered CRE orally for 2 weeks (50 mg/kg body weight). The vehicle group was treated with the same capacity of distilled water as the CRE treatment. The mice were sacrificed at 2 weeks, and the organs were isolated. Three mice were used for each vehicle and CRE group in the experiment.

### 2.3. Cell Culture

All cell culture reagents were purchased from Welgene (Gyungsan, Korea). Murine delayed brain tumor (DBT) cell was maintained at 37 °C in a 5% CO_2_ atmosphere in DMEM (Welgene, LM001-05) supplemented with 5% (vol/vol) fetal bovine serum and 1% penicillin-streptomycin (vol/vol) in six-well tissue culture plates. The experimental group was supplemented with CRE (2 μg, 4 μg/mL), Dichloroacetate acid (DCA) (20 mM), and 2-Deoxy-d-glucose (2-DG) (5 mM). All cell experiments were repeated at least 3 times.

### 2.4. Western Blotting

Both protein samples of cerebrums and DBT cells were extracted using a protein lysis buffer called T-PER reagent (78510, Thermo Fisher Scientific, Waltham, MA, USA) and quantified by Bradford assay with PRO-Measure solution (#21011, Intron, Kirkland WA, USA). Depending on the protein size, the samples were run using SDS-PAGE electrophoresis on 10 or 12% polyacrylamide gels and transferred to the membrane. The membranes were blocked for 1 h with 30 mg/mL BSA100 (9048-46-8, LPS solution, Daejeon, Korea), diluted TBS-T buffer (04870517TBST4021, LPS solution). Primary antibodies were operated overnight at 4 °C. Following this step, the membranes were washed with TBS-T, and the secondary antibodies were operated in an identical way. Results were detected with ECL solution (XLS025-0000, Cyanagen, Bologna, Italy) and Chemi Doc (Fusion Solo, VilberLourmat, Collégien, France). The primary antibodies were diluted at 1:2500 in 5% *w*/*v* BSA, and the secondary antibodies were diluted at 1:2500 in 5% *w*/*v* skim milk. The primary and secondary antibodies information is in Table 1.

### 2.5. Total RNA Extraction and Real-Time Quantitative PCR

Total RNA was extracted using TRIzol Reagent (15596-026, Life technologies, Carlsbad, CA, USA) in both mouse cerebrum and DBT cells. Reverse transcription was performed with 1.5 µg of total RNA and a Reverse transcriptase kit (SG-cDNAS100, Smartgene, Lausanne, Switzerland) following the manufacturer’s protocol. Quantitative PCR (real-time PCR) was executed using Excel Taq Q-PCR Master Mix (SG-SYBR-500, Smartgene) and Stratagene Mx3000P (Agilent Technologies, Santa Clara, CA, USA). The primers used in real-time PCR were manufactured by Bionics Inc. (Seoul, Korea) or Genotech (Seoul, Korea). RPLP0 was used as a control in in vivo and in vitro samples, respectively. All experiments were run more than in triplicate, and mRNA values were calculated based on the cycle threshold and monitored for an amplification curve. The primers used for real-time PCR are shown in Table 2. 

### 2.6. Measurements of Cellular Glycolysis

DBT cells were grown in DMEM, 10% FBS, and 1% penicillin/streptomycin media and incubated for 24 h. Additionally, after removing the media, CRE (4 μg/mL) was treated for 24 h. For the glycolysis stress test, the cells were further incubated in DMEM low-glucose medium [glucose 50 mg/dL without FBS]. Before the experiment, the cells were decarboxylated for 40 min to 1 h in an XFp medium (103575-100, Agilent Technologies) containing the same amount of glutamine, sodium pyruvate, and glucose as that of the medium in which the cells were grown. For the glycolysis stress test, glucose (25 mM) was used, and the extracellular acidification rate (ECAR) was measured. Seahorse XFp analyzer (Agilent Technologies), Seahorse XFp, XFp FluxPak (103022-100, Agilent Technologies), and an XFp Cell Mito Stress Test kit (103010-100, Agilent Technologies) were used for analysis.

### 2.7. Statistical Analysis

Data are reported as mean ± SEM. Differences between means were obtained by Student’s t-test and the one-way ANOVA followed by a Dunnett post-analysis performed using Graph Pad Software (GraphPad Inc., San Diego, CA, USA).

## 3. Results

### 3.1. Curcumae Radix Extract Reduced Markers of the ER Stress in Mouse Cerebrum

Twelve-week-old male C57BL/6 mice were treated with CRE for 30 days (Figure 1A), and their brain tissue was sampled and divided into several parts for analysis. We observed changes in the body weight and serum glucose levels of the mice to confirm basic metabolic changes (Figure 1B,C). Changes in body weight and serum glucose levels did not differ between the groups. To investigate whether neuronal damage was reduced by curcumin of CRE, we analyzed apoptosis and ER stress protein marker levels in mouse cerebrum. First, the apoptosis-related markers were estimated. The cleaved Caspase-3/Caspase-3 ratio and cleaved Poly (ADP-ribose) polymerase (PARP) levels showed no difference between the vehicle and CRE-treated groups (*p* < 0.05; Figure 1D). In addition, the Bcl-2-associated X protein (Bax)/B-cell lymphoma 2 (Bcl-2) ratio did not differ between the vehicle and CRE-treated groups (*p* < 0.05; Figure 1E). Next, ER stress-related markers showed that the phospho-Inositol-requiring transmembrane kinase endoribonuclease-1α (pIRE1α)/IRE1α ratio significantly (51%) decreased, Activating Transcription Factor 6 (ATF6) did not differ, and Chop significantly (76%) decreased between the vehicle and CRE-treated groups (*p* < 0.05; Figure 1F).

### 3.2. Curcumae Radix Extract Reduced Neurodegenerative Markers

We showed the ER stress decrease in the CRE-treated mouse brain. Since the neuroprotective effect could occur through ER stress decrease, we monitored the markers of dementia factor, a typical cause of ND development. The levels of App (85%), Aβ (77%), and tau protein (68%) were significantly lower in the CRE-treated group than in the vehicle group (*p* < 0.05; Figure 2A). CRE reduced neurodegenerative marker proteins in mouse cerebrum. Additionally, we treated the delayed brain tumor (DBT) cell line with CRE for 24 h and App levels significantly decreased in 2.5 µg/mL (65%) and 4 µg/mL (68%) CRE-treated cells (Figure 2B). Similar to the level of App protein, that of Aβ significantly decreased in 4 µg/mL (70%) CRE-treated cells, and that of Tau decreased in 2.5 µg/mL (75%) and 4 µg/mL (68%) CRE-treated cells (*p* < 0.05; Figure 2B). These results indicate that Curcumae radix can provide protection against ND.

### 3.3. Curcumae Radix Extract Decreased Glycolysis and Compensatively Increased the TCA Cycle in Mouse Cerebrum

Curcumae radix has anti-inflammatory and antioxidant properties and limits the aggregation of misfolded proteins in the brain. However, its role in energy metabolism is not clear [33]. Therefore, we focused on the relationship between Curcumae radix and energy metabolism and measured glycolysis-related protein levels. The protein level of hexokinase1 (HK1) was higher in the CRE-treated group than in the vehicle group (1.67-fold; Figure 3A). In contrast, the levels of pyruvate dehydrogenase (PDH; 65%) and pyruvate kinase M2 (PKM2; 82%) were lower, and those of lactate dehydrogenase A (LDHA; 1.6-fold) were higher in the CRE-treated group than in the vehicle group (*p* < 0.05; Figure 3A). However, we cannot suggest an increase or decrease in glycolysis as the relationship between HK1 and PKM2 was not definite. When we investigated the glucose transporter 1 glut(Glut1) mRNA level, Glut1 mRNA was decreased in the CRE-treated group (77%, *p* < 0.05; Appendix A). Finally, we investigated the changes in the TCA cycle and OXPHOS. TCA cycle-related mRNA levels were increased in Cs (3.29-fold), Aco2 (5.70-fold), Ogdh (3.72-fold), Idh3a (3.15-fold), Sdhb (2.56-fold), and Mdh2 (3.28-fold). Additionally, OXPHOS-related mRNA gene levels were increased (ATP5b: 2.24-fold, Ndufb5: 1.23-fold, and slc25a4: 1.54-fold) in the CRE-treated group compared to those in the vehicle group (*p* < 0.05; Figure 3B,C). 

### 3.4. Curcumae Radix Extract Decreased Glycolysis Markers and Compensatively Increased the TCA Cycle in DBT Cells

When glial cells promote glycolysis, they convert and transfer lactate to neurons [34]. As glial cells play a role in supporting neurons in metabolic aspects [35], glial-like DBT cells were introduced in this study. In a previous study, we showed that Curcumae radix changed glycolysis and increased the TCA cycle in the mouse cerebrum. To supplement this information, we conducted a cell experiment. First, we evaluated changes in the levels of neurodegenerative markers. Next, we measured the levels of glycolysis-related markers. HK1 significantly increased by 1.43-fold (*p* < 0.05; Figure 4A), whereas PDH and PKM2 significantly decreased in a dose-dependent manner (82% and 78%, respectively). Similar to HK1, LDHA significantly increased in 2.5 µg/mL (1.47-fold) and 4 µg/mL (1.62-fold) CRE-treated cells (*p* < 0.05; Figure 4A). Moreover, Glut1 mRNA decreased in the CRE-treated cells (70%, *p* < 0.05; Appendix A). As expected, the glycolysis data were similar in vivo and in vitro. To determine whether curcumin actually increases or decreases glycolysis, we evaluated cellular glycolysis (Extracellular acidification rate, ECAR) using a metabolic flux assay in the CRE-treated DBT cells. Surprisingly, glycolytic activity (ECAR) significantly decreased (53%) in the CRE-treated group compared to that in the vehicle group (*p* < 0.05; Figure 4B). Finally, we investigated the TCA cycle and OXPHOS in the DBT cell line treated with CRE for 24 h. TCA cycle-related mRNA levels were increased in Cs (1.7-fold), Aco2 (2-fold), Ogdh (1.71-fold), Suclg (2.11-fold), Sdhb (1.3-fold), and Mdh2 (1.56-fold) cells. Additionally, OXPHOS-related mRNA gene levels were increased (Ndufb5: 1.23-fold and slc25a4: 1.61-fold) in the CRE-treated group compared to those in the vehicle group (*p* < 0.05; Figure 4C,D). 

### 3.5. Neurodegenerative Markers Decreased in Glycolysis Inhibition and TCA Activation State of DBT Cells

We observed the inhibition of glycolysis and activation of the TCA cycle when CRE was added to DBT cells. First, we investigated whether ND markers decreased glycolysis inhibition and TCA activation in an in vitro assay. When DBT cells were treated with a glycolysis inhibitor (2-Deoxy-d-glucose, 2-DG) for 24 h, App (86%) and Tau (77%) levels significantly decreased in the 2-DG-treated group compared to those in the vehicle group (Figure 5A). When DBT cells were treated with a TCA activator (Dichloroacetate acid, DCA) for 24 h, App (58%), Aβ (57%), and Tau (77%) were significantly decreased in the DCA-treated group as compared to those in the vehicle group (*p* < 0.05; Figure 5B). In addition, the levels of glycolytic HK1 (64%) and PKM2 (81%) were significantly lower in the 2-DG-treated group as compared to those in the vehicle group (*p* < 0.05; Figure 5C). However, PDH and LDHA levels were similar to those in the vehicle group (Figure 5C). Glycolytic proteins did not differ between the vehicle and DCA-treated groups (Figure 5D). Next, we monitored the TCA cycle and OXPHOS-related mRNA levels in DBT cells treated with 2-DG or DCA. The TCA cycle genes Aco2 (62%), Ogdh (81%), and Mdh2 (82%) decreased and Idh3a (1.28-fold), Suclg (1.35-fold), and Sdhb (1.12-fold) increased in the 2-DG-treated DBT cells (*p* < 0.05; Figure 5E). While the OXPHOS Ndufb5 (1.41-fold) mRNA was increased in the 2-DG-treated group, slc25a4 (64%) decreased in the 2-DG-treated group as compared to that in the vehicle group (*p* < 0.05; Figure 5E). The TCA cycle genes Cs (1.68-fold), Aco2 (2.15-fold), Ogdh (2.48-fold), Suclg (1.4-fold), Sdhb (2.43-fold), and Mdh2 (1.71-fold) were increased in the DCA-treated group. OXPHOS ATP5b (1.44-fold) and slc25a4 (2.56-fold) also increased in the DCA-treated group as compared to those in the vehicle group (*p* < 0.05; Figure 5F).

### 3.6. Curcumae Radix Extracts Have Neuroprotective Effects in Tau-Overexpressing Mouse Cerebrum

The tau-overexpression mouse model is considered to be more representative of ND than other animal models. We analyzed the protein levels of the main factors identified as typical causes of Alzheimer’s disease initiation. App (84%) and tau proteins (71%) significantly decreased (*p* < 0.05) in the CRE-treated group as compared to those in the vehicle group, while the difference was not significant for Aβ (Figure 6B). Successively, the protein level of HK1 was increased (2.45-fold) in the CRE-treated group compared to that in the vehicle group. In contrast, the levels of PDH (67%) and PKM2 (88%) were lower, and those of LDHA (2.02-fold) were higher in the CRE-treated group than those in the vehicle group (*p* < 0.05; Figure 6C). In addition, Glut1 mRNA was decreased in the CRE-treated group (46%, *p* < 0.05; Appendix A). Finally, the TCA cycle was investigated. TCA cycle-related mRNA levels were increased in Cs (1.7-fold), Aco2 (2.29-fold), Ogdh (1.71-fold), Idh3a (1.33-fold), Suclg (1.88-fold), Sdhb (1.29-fold), and Mdh2 (1.27-fold) (*p* < 0.05; Figure 6D). 

## 4. Discussion

The incidence of ND has increased owing to an aging society. Several researchers have focused on the prevention and treatment of ND. Alzheimer’s disease is a major degenerative disease. However, treatment options are currently nonexistent, and the symptoms gradually progress, making continuous prevention important. 

Our previous study identified the components of CRE, such as curcumin, demethoxycurcumin, and bis-demethoxycurcumin, and assessed their anti-migratory activity in a breast cancer model [10]. Another study found that curcumin inhibited ER stress and apoptosis [36,37], demethoxycurcumin has neuroprotective effects [38,39], and bisdemethoxycurcumin inhibits ND by suppressing oxidative stress [40,41]. Therefore, we measured neurodegenerative, apoptotic, and ER stress markers in the mouse cerebrum. The results showed that Curcumae radix decreased the levels of neurodegenerative markers and ER stress. Though Curcumae radix was shown to decrease neurodegenerative markers [42,43,44], its role in affecting energy metabolism in ND remains unclear. Some studies investigated ER stress and energy metabolism in the heart [45,46]. Therefore, we focused on the role of Curcumae radix in energy metabolism in the brain.

In this study, the mouse cerebrum increased HK1, which is the first step in the phosphorylation of glucose to glucose-6-phosphate [47]. In contrast, PKM2, which catalyzes the last step of glycolysis and produces pyruvate [48], was decreased in the mouse brain. To confirm whether Curcumae radix is involved in the increase or decrease of glycolysis, we used ECAR, a real-time glycolysis metabolic monitoring system. We found that ECAR decreased in DBT cells treated with CRE, similar to a previous report in which curcumin reduced ECAR in gastric tumor cells [49], suggesting that Curcumae radix could decrease glycolysis in glial cells. Interestingly, while a low glycolysis rate was observed, the TCA cycle and OXPHOS mRNA expression levels were increased in cells treated with the Curcumae radix extract. Curcumin, a polyphenol, improves mitochondrial function [50] and activates lipid metabolism [51]. We expected that the TCA cycle function would be enhanced, which would reduce the use of glucose and glycolysis. As activation of the TCA cycle improved mitochondrial function during ND, TCA cycle activation in Curcumae radix extract-treated cells could improve mitochondrial function and delay neurodegenerative disease. Similar to TCA cycle activation, mitochondrial biogenesis is known to help retard the onset and progression of ND [52]. In fact, neurodegenerative markers decreased in the Curcumae radix-treated group with active mitochondrial function. We suggest that curcumin affects energy metabolism by activating the TCA cycle. 

As glycolysis is relatively reduced due to TCA cycle activation, unused glucose is stored as glycogen or used in the pentose phosphate pathway. Glucose is stored in the form of glycogen, primarily in astrocytes. Additionally, glycogen may support neuronal function when the glucose supply from the blood is poor [53] or during neuronal activation. In addition, the pentose phosphate pathway supplies NADPH for the removal of intracellular ROS [54]. Our results showed that HK1 was increased in the Curcumae radix-treated group. In a sufficient energy state, HK1 activation induces glucose-6-phosphate, which can be stored as glycogen through the action of glycogen synthase [55]. As a product of hexokinase, glucose-6-phosphate can induce the pentose phosphate pathway for NADPH production, which is involved in ROS reduction and ER stress [56]. It was reported that the antioxidant effect of curcumin is mediated by ROS removal through the activation of the high HK1-induced pentose phosphate pathway [57]. Similarly, neurodegenerative markers decreased with increasing HK1 expression in the Curcumae radix-treated group.

As Curcumae radix treatment decreased glycolysis and increased the TCA cycle in vivo and in vitro, we investigated neurodegenerative and energy metabolism in glycolysis inhibition or TCA cycle activation states. Interestingly, the levels of neurodegenerative markers decreased in glycolysis inhibitor (2-DG)-treated or TCA cycle activator (DCA)-treated DBT cells. Previous studies found that 2-DG maintained and promoted the mitochondrial bioenergetic pathway, indicating that ketone body metabolism can support neuronal function [58]. DCA is also known to enhance mitochondrial function and protect against ND [59]. Similar to the role of 2-DG and DCA, curcumin could activate lipid metabolism [51] and mitochondrial function [50], and ketones could be used instead of glucose in the brain [60]. Interestingly, some studies reported that ketone bodies or medium-chain fatty acids in the brain through mitochondrial respiration could be used as therapeutic agents for ND [61,62]. Our results showed that in Curcumae radix-treated cells, curcumin was used as an energy source together with glucose, and TCA was activated, indicating a sufficient energy state (Appendix A). Collectively, our results suggest that Curcumae radix affects the TCA cycle activator. 

Interestingly, recent research reported that tau interacts with proteins involved in mitochondrial bioenergetics and mutant tau links with mitochondrial protein atrophy and mitochondrial dysfunction [63]. Unlike App, tau protein is a component of neurofibrillary tangles and a microtubule-associated protein that revitalizes the polymerization of microtubules and may play a role in cell signaling [64]. In particular, in the early stages of Alzheimer’s disease, progressive memory problems are associated with the tau protein [65]. In the pathological conditions of Alzheimer’s disease, tau destabilizes microtubules when excessively phosphorylated [66]. Since curcumin enhances energy and mitochondrial metabolism [67,68], the TCA cycle activation states might be represented by the Curcumae radix treatment. We used hTau23 Tg mice (24 weeks old), which showed high levels of tau phosphorylation with advanced age [69]. As an efficient supplement to neuronal metabolism, Curcumae radix extracts decreased App and tau proteins through the activation of the TCA cycle in tau transgenic mice. Thus, Curcumae radix was considered to improve ND through TCA cycle activation.

## 5. Conclusions

Curcumae radix decreases glycolysis and activates the TCA cycle in the brain. The results of our study suggest that Curcumae radix acts as a TCA cycle activator and compensatively increases the TCA cycle by acting as a glycolysis inhibitor. As CRE could protect against Alzheimer’s disease by activating the TCA cycle of mitochondrial metabolism, we highlight its potential as a means for the treatment and prevention of neurodegenerative diseases.

## Figures and Tables

**Figure 1 nutrients-14-01587-f001:**
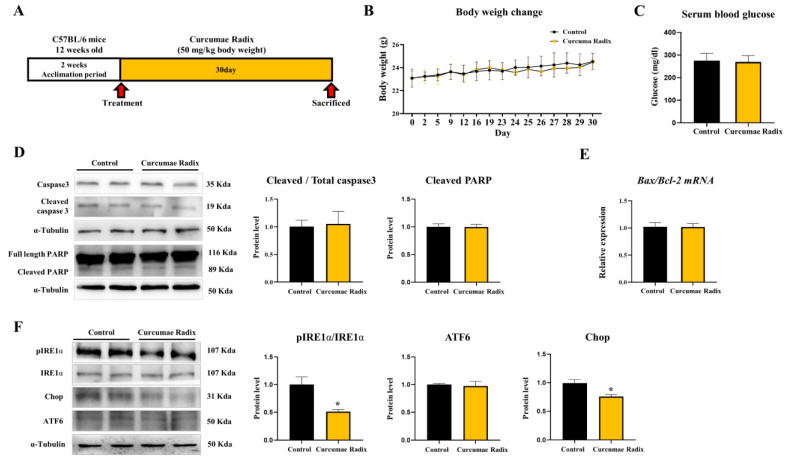
Curcumae radix decreases endoplasmic reticulum (ER) stress in mouse brain. (**A**) The Schematic diagram shows the schedule of the animal experiment. Twelve-week-old male mice had an acclimation period of 2 weeks. For 23 days five times a week, they were injected orally. For the last 7 consecutive days of administration, the CRE group was treated with Curcumae radix extract (CRE), and the vehicle group was treated with water (50 mg/kg body weight, *n* = 6). (**B**) Mouse body weight monitoring data. (**C**) Serum blood glucose level measured after sacrificed. (**D**) Western blot analysis and quantification of apoptosis-related genes were evaluated in the cerebrum of CRE male mice. Alpha-Tubulin was used as an internal control. (**E**) Bcl-2-associated X protein (Bax), B-cell lymphoma 2 (Bcl-2) genes mRNA levels were measured by quantitative RT-PCR and Bax, Bcl-2 genes mRNA levels were determined by the Bax/Bcl-2 ratio in the cerebrum of CRE treated male mice. RPLP0 was used as an internal control. (**F**) Western blot analysis and quantification of ER stress-related genes were evaluated in the cerebrum of CRE treated male mice. Alpha-Tubulin was used as an internal control. The values stand for means ± S.D. * *p* < 0.05 was compared to groups indicated. All experiments were repeated at least 3 times.

**Figure 2 nutrients-14-01587-f002:**
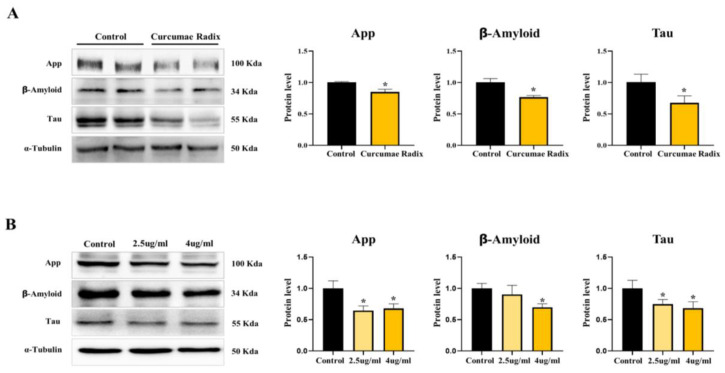
Curcumae radix reduces neurodegenerative markers. (**A**)Western blot analysis and quantification of Alzheimer’s disease-related genes were evaluated in the cerebrum of CRE-treated male mice. Alpha-Tubulin was used as an internal control. (**B**) Western blot analysis and quantification of Western blot analysis and quantification of Alzheimer’s disease-related genes were evaluated in delayed brain tumor (DBT) cells of CRE-treated mice. Alpha-Tubulin was used as an internal control. The CRE group was treated with CRE 2.5 μg/mL and 4 μg/mL. The values stand for means ± S.D. * *p* < 0.05 was compared to groups indicated. All experiments were repeated at least 3 times.

**Figure 3 nutrients-14-01587-f003:**
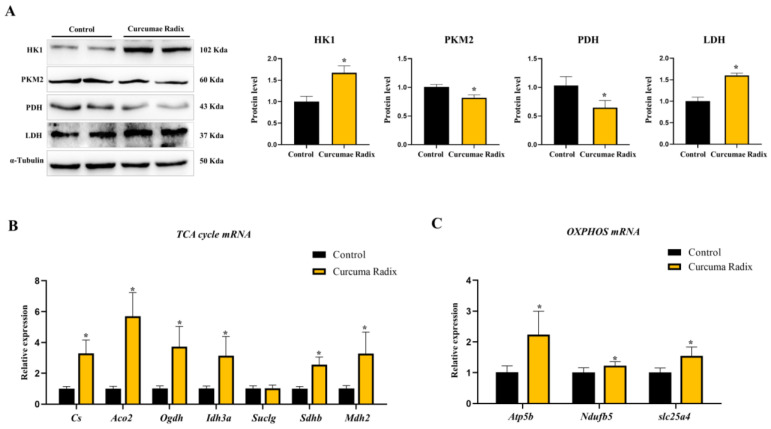
Curcumae radix regulates glycolysis and increases the Tricarboxylic acid (TCA) cycle, Oxidative phosphorylation (OXPHOS). (**A**) Western blot analysis and quantification of glycolysis related genes were evaluated in the cerebrum of CRE treated male mice. Alpha-Tubulin was used as an internal control. (**B**) TCA cycle genes mRNA levels were measured by quantitative RT-PCR in the cerebrum of CRE-treated male mice. RPLP0 was used as an internal control. (**C**) OXPHOS mRNA levels were measured by quantitative RT-PCR in the cerebrum of CRE-treated male mice. RPLP0 was used as an internal control. The values stand for means ± S.D. * *p* < 0.05 was compared to the groups indicated. All experiments were repeated at least 3 times.

**Figure 4 nutrients-14-01587-f004:**
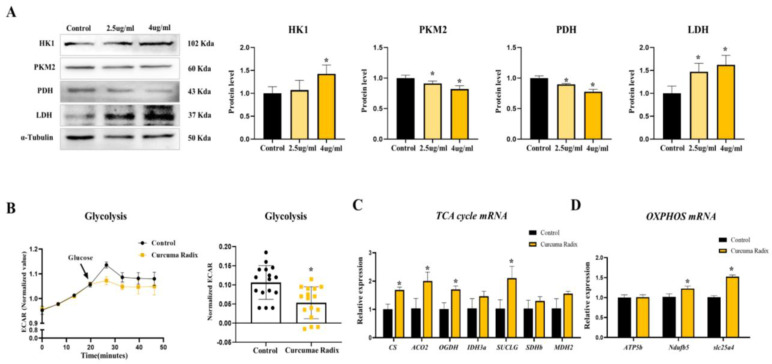
Curcumae radix decreases glycolysis and increases the TCA cycle. (**A**) Western blot analysis and quantification of glycolysis related genes were evaluated in DBT cells of CRE-treated mice. Alpha-Tubulin was used as an internal control. CRE group treated CRE 2.5 μg/mL,4 μg/mL. (**B**) Glycolysis rate measured using flux analyzer in DBT cells. Values were normalized to baseline. Glycolysis was evaluated by measurements from the extracellular acidification rate (ECAR). As indicated by the arrow, glucose was injected at a concentration of 25 mM in the cell. The CRE group was treated with CRE 4 μg/mL. (**C**) TCA cycle genes mRNA levels were measured by quantitative RT-PCR in DBT cells of CRE-treated mice. The CRE group was treated with CRE 4 μg/mL. RPLP0 was used as an internal control. (**D**) OXPHOS genes mRNA levels were measured by quantitative RT-PCR in DBT cells of CRE-treated mice. The CRE group was treated with CRE 4 μg/mL. RPLP0 was used as an internal control. The values stand for means ± S.D. * *p* < 0.05 was compared to the groups indicated. All experiments were repeated at least 3 times.

**Figure 5 nutrients-14-01587-f005:**
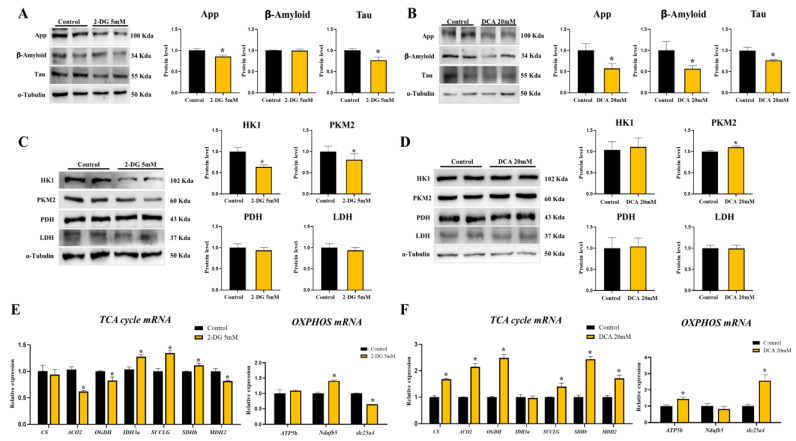
Glycolysis inhibition and TCA activation condition decrease neurodegenerative markers. (**A**) Western blot analysis and quantification of Western blot analysis and quantification of Alzheimer’s disease-related genes were evaluated in DBT cells of 2-Deoxy-d-glucose (2-DG) 5 mM-treated mice. Alpha-Tubulin was used as an internal control. (**B**) Western blot analysis and quantification of Western blot analysis and quantification of Alzheimer’s disease-related genes were evaluated in DBT cells of Dichloroacetate acid (DCA) 20 mM-treated mice. Alpha-Tubulin was used as an internal control. (**C**) Western blot analysis and quantification of glycolysis related genes were evaluated in DBT cells of 2-DG 5 mM-treated mice. Alpha-Tubulin was used as an internal control. (**D**) Western blot analysis and quantification of glycolysis related genes were evaluated in DBT cells of DCA 20 mM-treated mice. Alpha-Tubulin was used as an internal control. (**E**) TCA cycle and OXPHOS genes mRNA levels were measured by quantitative RT-PCR in DBT cells treated with 2-DG. RPLP0 was used as an internal control. (**F**) TCA cycle and OXPHOS genes mRNA levels were measured by quantitative RT-PCR in DBT cells treated with DCA. RPLP0 was used as an internal control. The values stand for means ± S.D. * *p* < 0.05 was compared to the groups indicated. All experiments were repeated at least 3 times.

**Figure 6 nutrients-14-01587-f006:**
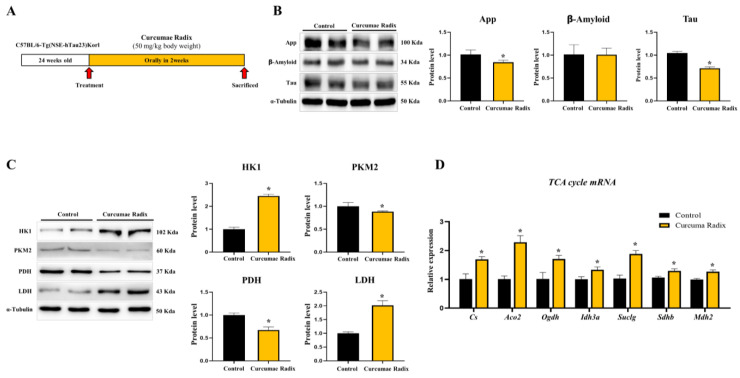
Curcumae radix decreases neurodegenerative markers through TCA cycle activation in Tau overexpression transgenic mice. (**A**) The Schematic diagram shows the schedule of the animal experiment. Twenty-four-week-old male mice were injected orally for 2 weeks. The CRE-treated group was treated with CRE, and the vehicle group was treated with water (50 mg/kg body weight, *n* = 3). (**B**) Western blot analysis and quantification of Alzheimer’s disease-related genes in the cerebrum of CRE treated Tau transgenic male mice. Alpha-Tubulin was used as an internal control. (**C**) Western blot analysis and quantification of glycolysis related genes were evaluated in the cerebrum of CRE treated Tau transgenic male mice. (**D**) TCA cycle genes mRNA levels were measured by quantitative RT-PCR in the cerebrum of CRE-treated Tau transgenic male mice. The CRE group was treated with CRE, and the vehicle group was treated with water (50 mg/kg body weight, *n* = 3). The values stand for means ± S.D. * *p* < 0.05 was compared to the groups indicated. All experiments were repeated at least 3 times.

**Table 1 nutrients-14-01587-t001:** Primary and secondary antibodies used for Western blot.

Primary Antibodies	Type	Lot.	Inc.
PRAP	Rabbit monoclonal	9532	Cell signaling technology
Phospho-IRE1α	Rabbit polyclonal	Ab37073	Abcam PLC
IRE1α	Rabbit polyclonal	Ab48187	Abcam PLC
Chop	Mouse monoclonal	MA1-250	Invitrogen
ATF6	Rabbit polyclonal	Ab65838	Abcam PLC
Beta-amyloid-	Mouse monoclonal	sc-28365	Santa Cruz biotechology
Tau	Rabbit monoclonal	A1103	Company ABclonal, Inc.
HK1ǀ	Rabbit monoclonal	2024	Cell signaling technology
PKM2	Rabbit monoclonal	4053	Cell signaling technology
PDH	Rabbit monoclonal	3205	Cell signaling technology
LDHA	Rabbit monoclonal	3582	Cell signaling technology
Alpha-Tubulin	Mouse monoclonal	66031-1-Ig	Proteintech Group Inc
AMPKα	Rabbit monoclonal	5831	Cell signaling technology
Phospho-AMPKα	Rabbit monoclonal	2535	Cell signaling technology
**Secondary antibodies**	**Type**	**Lot.**	**Inc.**
Anti-Mouse IgG	Goat	121507	Jackonimmuno
Anti-Rabbit IgG	Mouse	123213	Jackonimmuno
IRE1α	Rabbit polyclonal	Ab48187	Abcam PLC

**Table 2 nutrients-14-01587-t002:** Primers used for real-time PCR.

Gene	Upper Primer (5′-3′)	Lower Primer (5′-3′)	Species
*Bax*	TGA AGA CAG GGG CCT TTT TG	AAT TCG CCG GAG ACA CTC	Mouse
*Bcl-2*	ATG CCT TTG TGG AAC TAT ATG GC	GGT ATG CAC CCA GAG TGA TGC	Mouse
*Cs*	CCT GAG TGC CAG AAA ATG CTG	CCA CAT GAG AAG GCA GAG CT	Mouse
*Aco2*	ACA AGT GGG ACG GCA AAG AC	AGC ATT GCG TAC AGA GTT GGC	Mouse
*Ogdh*	AAT GCT GAG CTG GCC TGG TG	TCA GGT GTG TTT TCT TGT TGC C	Mouse
*Idh3a*	TGC TTC GCC ACA TGG GAC TT	CGT TGC CTC CCA GAT CTT TT	Mouse
*Suclg2*	CTG TGC CAT CAT TGC CAA CG	ATG GGG AGT CCG CTG CTC TT	Mouse
*Sdhb*	CTC TGT CTA CCG CTG CCA C	GGC ACA CTC AGC ACG GAC T	Mouse
*Mdh2*	ATG CTG GAG CCC GCT TTG TC	CAG GGA TAG CCT CGG CAA TC	Mouse
*Atp5b*	CCC TGA AGG AGA CCA TCA AA	AAG ACC CCT CAC GAT GAA TG	Mouse
*Ndufb5*	CTT CCT CAC TCG TGG CTT TC	CGC ACT TCC AGC TCC TTT AC	Mouse
*Slc25a4*	ATG GTC TGG GCG ACT GTA TC	TCA AAG GGG TAG GAC ACC AG	Mouse
*RPLP* *0*	GCA GCA GAT CCG CAT GTC GCT CCG	GAG CTG GCA CAG TGA CCT CAC ACG G	Mouse

## Data Availability

Not applicable.

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
