# Peer review of "Curcumae Radix Decreases Neurodegenerative Markers through Glycolysis Decrease and TCA Cycle Activation"

_nutrients, 2022, doi:10.3390/nu14081587_

Round 1
Reviewer 1 Report
In this manuscript Hong and colleagues investigated the relationship between neurodegenerative diseases and energy metabolism using Curcumae radix extract in mice and murine delayed brain tumor (DBT) cells model. They found that levels of Alzheimer’s disease-related markers in mouse brains were reduced after treatment with the Curcumae radix extract and neurodegenerative markers decreased in the cerebrum and brain tumor cells following low ER stress markers. In addition, Curcumae radix extract decreased glycolysis-related genes and the extracellular acidification rate. Moreover, Curcumae radix-exposed mouse brain and cells had increased mitochondrial TCA cycle genes and OXPHOS-related genes suggesting that curcumae radix may act as a metabolic modulator of brain health and help treat and prevent neurodegenerative diseases.
Although it is a very interesting study, many points need to be improved:
- the reader suggest to abbreviate Curcumae radix extract with "CRE" in the text
- in the introduction section, athors should also point out that curcumin also inhibits STAT3 activation in human adipocytes (PMID: 31781039), a key signaling in insulin resistance onset (PMID: 32101031). Moreover, curcumin plays also a key role in obesity and gestational diabetes (PMID: 34981472, 33477354). All conditions that affect glucose metabolism.
- A list of abbreviations with relative full name must be shown
- Line 60-64: references must be formatted according to the journal style
- Line 118: "on 10, 12% polyacrylamide gels". 10%, 12% or both?
- Line 119: How long membranes have been blocked with BSA?
- Table 1: the antibody dilutions must be reported
- Line 161: it is not clear, syntax must be checked out
- Line 320-322: "This section may be divided by subheadings. It should provide a concise and precise description of the experimental results, their interpretation, as well as the experimental conclusions that can be drawn." It must be deleted since is part of the journal template (Line 53-55 of template file)
- In order to further validate the results obtained, glucose uptake assay should also be performed
- An accurate revision of typing errors and syntax is recommended
Author Response
- the reader suggest to abbreviate Curcumae radix extract with "CRE" in the text
- According to the reviewer’s suggestion, we abbreviated Curcumae radix extract with “CRE” in the abstract and introduction.
- in the introduction section, authors should also point out that curcumin also inhibits STAT3 activation in human adipocytes (PMID: 31781039), a key signaling in insulin resistance onset (PMID: 32101031). Moreover, curcumin also plays a key role in obesity and gestational diabetes (PMID: 34981472, 33477354). All conditions that affect glucose metabolism.
-According to the reviewer’s suggestion, we added and revised the introduction.
Previous studies have found that curcumin stimulates glucose uptake and activates glucose metabolism in skeletal muscles [23,24]. However, it also reduces the expression levels of glycolysis-related enzymes and lactate dehydrogenase A in the liver [25-28]. In diabetes study, curcumin inhibited adipose STAT3 activation, essential signaling in insulin resistance onset [29, 30]. Moreover, recent studies reported that curcumin affected protection against obesity and gestational diabetes [31,32]. Surprisingly, when treated with Curcumae radix, glycolysis metabolism and neurodegenerative markers decreased in the mouse brain and neurodegenerative disease mod-el-tau overexpression mouse brain. Though CRE has been extensively studied, its role in energy metabolism in neurodegenerative diseases remains unclear. In this study, we focused on the effects of CRE on neurodegenerative diseases and energy metabolism.
- A list of abbreviations with relative full name must be shown
- According to the reviewer’s suggestion, we revised the abbreviations following journal format.
- Line 60-64: references must be formatted according to the journal style
- According to the reviewer’s suggestion, we unified the reference style.
As demonstrated in several studies, the effects of curcumin are linked to its an-ti-inflammatory [13,14], antimicrobial [15], and anticancer properties [10,16-18] and, in the brain, curcumin was known to inhibited endoplasmic reticulum stress (ER stress) and apoptosis [19-22].
- Line 118: "on 10, 12% polyacrylamide gels". 10%, 12% or both?
- Depending on the protein size, we used 10% or 12% polyacrylamide gels.
- Line 119: How long membranes have been blocked with BSA?
-After transfer, the membrane was incubated with 30mg/ml BSA blocking buffer for 1hr.
Depending on the protein size, the samples were run SDS-PAGE electrophoresis on 10 or 12% polyacrylamide gels and transferred to the membrane. The membranes were blocked for 1hr with 30mg/ml BSA100 (9048-46-8, LPS solution), diluted TBS-T buffer (04870517TBST4021, LPS solution).
- Table 1: the antibody dilutions must be reported
- According to the reviewer’s comment, we described the antibody dilutions in the “Materials and Methods” (2.4 Western blotting) paragraph.
The primary antibodies diluted 1:2500 in 5% w/v BSA, and secondary antibodies diluted 1:2500 in 5% w/v skim milk. (Line 123-124)
- Line 161: it is not clear, syntax must be checked out
-According to reviewer’s opinion, we revised the sentence.
Changes in body weight and serum glucose levels did not differ between the groups. To investigate whether neuronal damage was reduced by curcumin of CRE, we analyzed apoptosis and ER stress protein marker levels in mouse cerebrum. (Line 161-163)
- Line 320-322: "This section may be divided by subheadings. It should provide a concise and precise description of the experimental results, their interpretation, as well as the experimental conclusions that can be drawn." It must be deleted since is part of the journal template (Line 53-55 of template file)
-According to reviewer’s opinion, we deleted the template.
- In order to further validate the results obtained, glucose uptake assay should also be performed
-According to the reviewer’s suggestion, we tried to measure glucose channel transcript in the brain instead of glucose uptake. As expected, Glut1 mRNA was decreased in the Curcumae radix extract group.
Figure S2. Curcumae Radix reduce glucose transport. (A) Glut1 mRNA levels were measured by quantitative RT-PCR in the cerebrum of CRE group treated CRE and vehicle group treated water (50 mg/kg body weight, n=6). (B) Glut1 mRNA levels were measured by quantitative RT-PCR in DBT cell of CRE treated. CRE group treated CRE 4ug/ml. (C) Glut1 mRNA levels were measured by quantitative RT-PCR in the cerebrum of CRE treated Tau transgenic male mice. CRE group treated CRE and vehicle group treated water (50 mg/kg body weight, n=3). RPLPO was used for an internal control. The values stand for means +/- S.D. *P<0.05 was compared to groups indicated. All experiments were repeated at least 3.
3.3 Curcumae radix extract decreased glycolysis and compensatively increased TCA cycle in mouse cerebrum
Curcumae radix has anti-inflammatory and antioxidant properties and limits aggregation of misfolded proteins in the brain. However, its role in energy metabolism is not clear [33]. Therefore, we focused on the relationship between Curcumae radix and energy metabolism and measured glycolysis-related protein levels. The protein level of hexokinase1 (HK1) was higher in the CRE-treated group than in the vehicle group (1.67-fold; Fig. 3A). In contrast, the levels of pyruvate dehydrogenase (PDH; 65%) and pyruvate kinase M2 (PKM2; 82%) were lower, and those of lactate dehydrogenase A (LDHA; 1.6-fold) were higher in the CRE-treated group than in the vehicle group (p < 0.05; Fig. 3A). However, we cannot suggest an increase or decrease in glycolysis as the relationship between HK1 and PKM2 was not definite. When we investigated glucose transporter 1 glut(Glut1) mRNA level, Glut1 mRNA was decreased in the CRE group treated group (77%, p < 0.05; Fig. S2A). Finally, we investigated the changes in the TCA cycle and OXPHOS. TCA cycle-related mRNA levels were increased in Cs (3,29-fold), Aco2 (5.70-fold), Ogdh (3.72-fold), Idh3a (3.15-fold), Sdhb (2.56-fold), and Mdh2 (3.28-fold). Also, OXPHOS related mRNA gene levels were increased (ATP5b: 2.24-fold Ndufb5: 1.23-fold and slc25a4: 1.54-fold) in the CRE-treated group compared to those in the vehicle group (p < 0.05; Fig. 3B, C).
3.4 Curcumae radix extract decreased glycolysis markers and compensatively increased TCA cycle in delayed brain tumor (DBT) cell
When glial cells promote glycolysis, they convert and transfer lactate to neurons [34]. As glial cells play a role in supporting neurons in metabolic aspects [35], glial-like DBT cells were introduced in this study. In a previous study, we showed that Curcumae radix changed glycolysis and increased the TCA cycle in the mouse cerebrum. To supplement this information, we conducted a cell experiment. First, we evaluated changes in the levels of neurodegenerative markers. Next, we measured the levels of glycolysis-related markers. HK1 significantly increased by 1.43-fold (p < 0.05; Fig. 4A) whereas PDH and PKM2 significantly decreased in a dose-dependent manner (82% and 78%, respectively). Similar to HK1, LDH significantly increased in 2.5 µg/mL (1.47-fold) and 4 µg/mL (1.62-fold) CRE-treated cells (p < 0.05; Fig. 4A). Moreover, Glut1 mRNA was decreased in the CRE-treated cells (70%, p < 0.05; Fig. S2B). As expected, the glycolysis data were similar in vivo and in vitro. To determine whether curcumin actually increases or decreases glycolysis, we evaluated cellular glycolysis (extracellular acidification rate, ECAR) using a metabolic flux assay in CRE treated DBT cells. Surprisingly, glycolytic activity (ECAR) significantly decreased (53%) in the Curcumae radix group compared to that in the vehicle group (p < 0.05; Fig. 4B). Finally, we investigated the TCA cycle and OXPHOS in the DBT cell line treated with CRE for 24 h. TCA cycle-related mRNA levels were increased in Cs (1.7-fold), Aco2 (2-fold), Ogdh (1.71-fold), Suclg (2.11-fold), Sdhb (1.3-fold), and Mdh2 (1.56-fold) cells. Also, OXPHOS related mRNA gene levels were increased (Ndufb5: 1.23-fold and slc25a4: 1.61-fold) in the Curcumae radix group compared to those in the vehicle group (p < 0.05; Fig. 4C, D).
3.6 Curcumae radix extracts have neuroprotective effects in tau-overexpressing mouse cerebrum
The tau-overexpression mouse model is considered to be more representative of neurodegenerative diseases than other animal models. We analyzed the protein levels of the main factors identified as typical causes of Alzheimer’s disease initiation. App (84%) and tau proteins (71%) significantly decreased (p < 0.05) in the Curcumae radix-treated group as compared to those in the vehicle group, while the difference was not significant for Aβ (Fig. 6B). Successively, the protein level of HK1 was increased (2.45-fold) in the Curcumae radix group than that in the vehicle group. In contrast, the levels of PDH (67%) and PKM2 (88%) were lower, and those of LDH (2.02-fold) were higher in the Curcumae radix group than those in the vehicle group (p < 0.05; Fig. 6C). In addition, Glut1 mRNA was decreased in the CRE-treated group (46%, p < 0.05; Fig. S2C). Finally, the TCA cycle was investigated. TCA cycle-related mRNA levels were increased in Cs (1.7-fold), Aco2 (2.29-fold), Ogdh (1.71-fold), Idh3a (1.33-fold), Suclg (1.88-fold), Sdhb (1.29-fold), and Mdh2 (1.27-fold) (p < 0.05; Fig. 6D).
- An accurate revision of typing errors and syntax is recommended
-According to reviewer’s opinion, we revised this manuscript.
"Please see the attachment"

Reviewer 2 Report
The paper entitled ‘Curcumae radix decreases neurodegenerative markers through glycolysis decrease and TCA cycle activation’ is aimed to detail characterization of Curcumae radix impact on neurodegeneration. The paper is interesting and informative. The following suggestions should be considered:
- Abstract: Informative but too long. The abstract should be a total of about 200 words maximum.
- Introduction: consistent and informative. Please unify reference style.
- Methods: authors decided to use various methods (in vivo, ex vivo, in vitro) which are adequate for the aim of the study. The methodology is well described and facilitate repetition of the experiments.
- Results: presented in form of tables and figures. Authors presented all necessary study results which are well described. Nevertheless Figures 1. and 5. are hard to analyse. Please improve the figures along with resolution improvement.
- Discussion: adequate to obtained study results. References are up-to-date and adequate.
- Conclusions: informative and consistent.
Author Response
Reviewer 2
The paper entitled ‘Curcumae radix decreases neurodegenerative markers through glycolysis decrease and TCA cycle activation’ is aimed to detail characterization of Curcumae radix impact on neurodegeneration. The paper is interesting and informative. The following suggestions should be considered:
1. Abstract: Informative but too long. The abstract should be a total of about 200 words maximum.
- According to the reviewer’s suggestion, we revised the abstract to 200 words maximum.
Neurodegenerative diseases (ND) are being increasingly studied owing to the increasing proportion of the aging population. Several potential compounds have been examined to prevent neurodegenerative diseases, including Curcumae radix, which is known to be beneficial in inflammatory conditions, metabolic syndrome, and various types of pain. However, it is not well studied and its influence on energy metabolism in ND is unclear. We focused on the relationship between ND and energy metabolism using Curcumae radix extract (CRE) in cells and animal models. We monitored neurodegenerative markers and metabolic indicators using western blotting and qRT-PCR, and then assessed cellular glycolysis and metabolic flux assays. Levels of Alzheimer’s disease-related markers in mouse brains were reduced after treatment with the CRE. We confirmed that neurodegenerative markers decreased in the cerebrum and brain tumor cells following low endoplasmic reticulum (ER) stress markers. Furthermore, glycolysis-related genes and the extracellular acidification rate decreased after treatment with the CRE. Interestingly, we found that the CRE exposed mouse brain and cells had increased mitochondrial Tricarboxylic acid (TCA) cycle and Oxidative phosphorylation (OXPHOS) related genes in the CRE group. Curcumae radix may act as a metabolic modulator of brain health and help treat and prevent ND involving mitochondrial dysfunction.
- Introduction: consistent and informative. Please unify reference style.
- According to the reviewer’s suggestion, we unified the reference style.
As demonstrated in several studies, the effects of curcumin are linked to its an-ti-inflammatory [13,14], antimicrobial [15], and anticancer properties [10,16-18] and, in the brain, curcumin was known to inhibited endoplasmic reticulum stress (ER stress) and apoptosis [19-22].
- Methods: authors decided to use various methods (in vivo, ex vivo, in vitro) which are adequate for the aim of the study. The methodology is well described and facilitate repetition of the experiments.
We appreciate the reviewer’s opinion.
- Results: presented in form of tables and figures. Authors presented all necessary study results which are well described. Nevertheless Figures 1. and 5. are hard to analyse. Please improve the figures along with resolution improvement.
- According to the reviewer’s suggestion, we improved all figures with high resolution.
- Discussion: adequate to obtained study results. References are up-to-date and adequate.
We appreciate the reviewer’s opinion.
- Conclusions: informative and consistent.
-We appreciate the reviewer’s opinion.
"Please see the attachment."

Round 2
Reviewer 1 Report
The manuscript has been significantly improved and can be accepted for publication in the present form.